# Quantitative Assessment of Asbestos Fibers in Normal and Pathological Pleural Tissue—A Scoping Review

**DOI:** 10.3390/life12020296

**Published:** 2022-02-16

**Authors:** Yohama Caraballo-Arias, Paola Caffaro, Paolo Boffetta, Francesco Saverio Violante

**Affiliations:** 1Department of Medical and Surgical Sciences, Alma Mater Studiorum University of Bologna, 40126 Bologna, Italy; yohama.caraballo@unibo.it (Y.C.-A.); francesco.violante@unibo.it (F.S.V.); 2School of Occupational Medicine, Alma Mater Studiorum University of Bologna, 40126 Bologna, Italy; paola.caffaro@studio.unibo.it; 3Stony Brook Cancer Center, Stony Brook University, New York, NY 11794, USA; 4Department of Medical and Surgical Sciences, IRCCS Azienda Ospedaliero-Universitaria di Bologna, 40138 Bologna, Italy

**Keywords:** asbestos fibers, pleural mesothelioma, electron microscopy, occupational diseases, chrysotile, amphiboles, scoping review

## Abstract

Background: pleural mesothelioma is a rare cancer in the general population, but it is more common in subjects occupationally exposed to asbestos. Studies with asbestos fiber quantification in pleural tissue are scarce: for this reason, we aimed at undertaking a scoping review to summarize the evidence provided by studies in which asbestos fibers were determined by electron microscopy (SEM or TEM) in human pleural tissues, whether normal or pathologic. Materials and methods: A scoping review of articles that quantified asbestos fibers in human pleural tissue (normal or pathologic) by electron microscopy (SEM or TEM), in subjects with asbestos exposure (if any) was performed. Results: The 12 studies selected comprised 137 cases, out of which 142 samples were analyzed. Asbestos fibers were detected in 111 samples (78%) and were below the detectable limit in 31 samples (22%). The concentration of asbestos fibers detected in the positive samples was distributed from as low as 0.01 mfgdt (millions of fibers per gram of dry tissue) up to 240 mfgdt. However, the minimum concentration of fibers overlaps in the three types of tissues (normal pleura, pleural plaque, mesothelioma) in terms of magnitude; therefore, it is not possible to distinguish a definite pattern which differentiates one tissue from the other. Conclusions: The studies included were heterogeneous as to the representativeness of the samples and analytical techniques; the possibility of false negatives must be considered. It would be desirable to systematically search for asbestos fibers to fill the knowledge gap about the presence of asbestos fibers in normal or pathological pleural tissue in order to better understand the development of the different pleural diseases induced by this mineral.

## 1. Introduction

Asbestos is a carcinogenic mineral and is the primary risk factor for malignant mesothelioma in the pleura, peritoneum, and other mesothelial tissues [1], in addition to being a risk factor for lung cancer and other diseases. Pleural mesothelioma is a rare disease in people not exposed to asbestos, with an estimated incidence of 1–2 per million per year [2]; however, in cohorts of subjects occupationally exposed to asbestos, up to 10% of deaths may be caused by malignant mesothelioma [3].

Asbestos fibers, especially amphiboles, have a long persistence in tissues and their presence has been studied for almost 50 years in humans [4], especially with electron microscopy (scanning-SEM or Transmission-TEM) [5].

Studies in which asbestos fibers have been determined in pleural tissues are relatively scarce, whereas the scientific literature has plentiful studies in which asbestos fibers have been determined in lung tissue. In addition, determination of asbestos fibers in the pleural tissues seems not to be a routine procedure even where pleural biopsies are performed for the diagnosis of pleural mesothelioma or other asbestos-related diseases.

For this reason, we aimed at undertaking a scoping review to summarize the evidence provided by studies in which asbestos fibers were determined by electron microscopy (SEM or TEM) in human pleural tissues, whether normal or pathologic.

## 2. Materials and Methods

The inclusion criteria for this review were:-articles written in any language, regardless of the publication date;-articles reporting a quantification of asbestos fibers in human pleural tissue (whether normal or pathologic) in subjects with defined or undefined asbestos exposure.

The exclusion criteria were:-articles not reporting a quantitative measure of the number of asbestos fibers found in human pleural tissue;-studies in non-human subjects or in tissues other than pleura.

Initially, the information source used for this review was PubMed using the terms “asbestos AND pleura AND electron*” on 30 September 2021.

The PubMed search yielded 184 articles that were independently evaluated by two of the authors (FSV, PC); 57 articles were discarded after reading the title, 58 after reading the abstract, and 59 after checking the full text. The references of each relevant article were manually searched, yielding two more papers; finally we identified 12 studies that fit our inclusion criteria [6,7,8,9,10,11,12,13,14,15,16,17].

Another search was conducted on Scopus, Compendex, Embase, GeoBase and Medline, which were searched using strings equivalent to the first one used in PubMed. The search of these other databases yielded 254 articles that were evaluated but did not fit the inclusion criteria.

To check for any overlooked but relevant articles, an additional search was later conducted on PubMed using the more inclusive string “asbestos AND pleura* AND electron*” which yielded 508 references, but none of the additional references retrieved applied to our purpose.

The 12 studies included were published between December 1974 and February 2001. Only 5 articles were published in the 1970s and 1980s [6,7,8,9,10]; most of the articles (7 papers) were published in the 1990s [11,12,13,14,15,16,17]. Remarkably, we could not locate any article fitting our inclusion criteria published in the last 20 years (from 2001 to 2021).

## 3. Results

Table 1 reports a description of the patients and conditions comprised in the studies included in this scoping review. Table 2 reports the number of asbestos fibers found in each tissue; studies are listed in chronological order of publication.

Le Bouffant [6] described a method for isolating asbestos fibers in biological tissues. The author analyzed the concentration of asbestos fibers in 12 samples (pleural and lung tissue) from 5 cases (no gender information provided), affected by pleural mesothelioma and pleural fibrosis; the subjects were asbestos workers. The minimum value of asbestos fibers detected in the pleural samples is not specified; the lowest concentration of asbestos fibers reported (a mesothelioma case) was 1.2 × 10^6^ fibers/gram of dry tissue. The author noted that, in cases of workers exposed to both types of asbestos fibers, amphiboles were prevalent in the lung parenchyma whereas chrysotile was prevalent in the pleura.

Sebastien et al. [7] assessed the concentration of asbestos fibers in 112 samples from 29 cases, affected by one or more conditions, such as lung fibrosis, bronchogenic carcinoma, pleural fibrosis, pleural effusions, and determined or suspected pleural mesothelioma, which were sent to their laboratory for confirmations of diagnosis. Five cases were female. No information is given whether “suspected pleural mesothelioma” was later confirmed as mesothelioma. In two cases, the occupation was “unknown”, and in three cases, asbestos exposure was “undiscovered”. The authors do not specify the detection limit of asbestos fibers associated with their techniques: however, according to Figure 1 of their paper [7], it seems to be in the range of 10^4^ × cc of tissue. They also report the results for cc of fixed tissue, which we approximated to 1 g of wet tissue. Almost all the fibers encountered in pleural tissues were of the chrysotile type.

Churg [8] established the concentration of asbestos fibers in 220 samples (mostly lung tissue) from 54 cases. The authors divided the cases into two groups: 29 subjects formed the plaque group and 25 subjects made up the control group. The plaque group included cases with pleural plaques observed at autopsy. The control group included 11 cases which had a very low number of asbestos fibers detected (less than 100 bodies/gram of wet lung tissue) and a lack of a history of asbestos exposure, and 14 cases without regard to asbestos exposure history or asbestos body count. Only one female was in the plaque group, and only one female was in the control group. The authors analyzed, for each case, four samples from lung tissue; only in four cases were pleural plaques also analyzed. According to the laboratory procedures, the authors set the detection limit for chrysotile at 10^4^ fibers/gram for wet lung tissue, whereas according to Table 6 of their paper [8], the minimum concentration of chrysotile in pleural plaques was 3 × 10^3^ fibers/gram of wet tissue. All the fibers encountered in four samples of pleural plaques were chrysotile.

Warnock et al. [9] reported the concentration of asbestos fibers in 124 samples (mostly lung tissue) from 31 cases affected by one or more conditions, such as pneumonia, leukemia, and other pathologies, except mesothelioma or macroscopic fibrosis. The study compares 20 subjects representing the plaque group (pleural plaques were observed at the autopsy) and 11 subjects representing the control group (with asbestos bodies less than 100/g of wet lung tissue). Only one female is included in the plaque group. The occupation in the first group is known, but the asbestos exposure is not clear. Three out of the seven cases of pleural samples where asbestos fibers were not detected had no clear history of asbestos exposure; these cases had, on average, had a much lower count of asbestos bodies in the lungs (364 per gram of wet lung tissue) than the five cases where asbestos was found in the plaque (4504 per gram of wet lung tissue). No information is available for the control group. The largest part of the samples analyzed were from the lung, and only 12 were from the pleura. The authors described the laboratory procedures but did not report a detection limit. However, as reported in the paper, the lowest concentration of asbestos fibers seems to be in the pleural tissue, with a value of 16 × 10^3^ fibers/gram of wet tissue. Both chrysotile and amphiboles fibers were detected in the samples.

Dodson et al. [10] assessed the concentration of asbestos fibers in eight samples from eight cases, affected by one or more conditions, such as asbestosis, lung cancer, laryngeal cancer, lung fibrosis, and emphysema. The information was obtained by clinical history and in four cases, asbestosis was diagnosed histologically. Only one case was female. The asbestos exposure was occupational, as all the subjects were former shipyard workers. According to the counting procedures of the fibers in each sample, the average detection limit was 41,000 fibers per gram of wet tissue and 200,000 per gram of dry tissue (lung and pleural tissues). Both chrysotile and amphiboles were detected in the samples.

Gibbs et al. [11] assessed the concentration of asbestos fibers in 13 cases, affected by different grades of diffuse pleural fibrosis. All subjects were men with an occupational history of asbestos exposure from 1 to 35 years. According to the authors, asbestos fibers counting encountered extreme values in case number 6, so they excluded it as an outlier. The authors did not report a detection limit; the minimum value of asbestos fibers in pleural tissue reported in Table 2 of their paper is 1.96 × 10^6^ fibers/gram of dry tissue (total asbestos fibers). Amphiboles fibers concentration was lower than chrysotile in pleura.

Kohyama & Suzuki [12] assessed the concentration of asbestos fibers in 33 samples from 13 cases affected by 1 or more conditions, such as asbestosis, lung cancer, and pleural and peritoneal mesothelioma, diagnoses which were validated by 1 of the authors. The study also analyzes asbestos fibers in two cases with diagnosis of peritoneal mesothelioma in which no pleural tissue was analyzed. All 13 cases were male. The asbestos exposure was specified in each case, with a duration from 27 up to 53 years. The authors report the results of the analyzed tissue (approximately 1 cc in volume), considering the weight to be about 0.1–0.2 g in their dry state. The lowest detection limit of asbestos fibers in pleural plaque tissue (Table 2 of the original study) was 0.16 × 10^6^ fibers/gram of dry tissue. Most of the fibers encountered in pleural tissues were of the chrysotile type, whereas the level of amphiboles was lower.

Paoletti et al. [13] assessed the concentration of asbestos fibers in 21 pleural samples from 21 cases, affected by pleural mesothelioma, diagnosed according to the laboratory procedures. Seven cases were female. In six cases, the occupation was “unknown”, and in five additional cases, asbestos exposure was “unlikely”. The authors report a variable detection limit (approximately down to 10^4^ fibers per gram of dry tissue) calculated according to Huang et al. [18]. Both chrysotile and amphiboles fibers were found in 12 positive cases: however, in 10 cases, only one type of fiber was present (in 6 cases, amphiboles, in 4 cases, chrysotile). In the 9 samples in which fibers were not detected, the authors state that the detection limit was generally high due to the small quantity of pleural tissue collected (less than 20 mg of dry tissue) and in the asbestos positive samples, the fiber count was low, frequently near the detection limit.

Tossavainen et al. [14] assessed the concentration of asbestos fibers in 10 samples from 10 cases, affected by 1 or more conditions, such as lung cancer, asbestosis, peritoneal mesothelioma, and pleural mesothelioma. Information about the duration and time of exposure was obtained from personal interviews, conducted by the authors. All the cases were men, and they were exposed to asbestos because of their occupations, with a duration of exposure ranging from 3 months up to 32 years. According to their laboratory analysis, the authors fixed the detection limit at 0.1 × 10^6^ fibers/gram of dry tissue. The authors reported the presence of amphiboles in the lung samples, and they encountered the same fibers analyzing various tissues (pleura, hilar lymph node, kidney cortex) of an asbestos sprayer.

Boutin et al. [15] studied the concentration of asbestos fibers in 14 samples from 14 cases, affected by 1 or more diseases, such as pleural mesothelioma, lung cancer, pleural effusion, and pneumothorax. The three diagnoses of mesothelioma were confirmed by the panel of pathologists of the French Mesothelioma Register. Two cases were female. In six cases, asbestos exposure was “occupational”, in one case “environmental”, in another case both “occupational and environmental”, and in six cases, no exposure was found. The laboratory analysis established a theoretical detection limit around 0.2 × 10^6^ fibers/gram of dry tissue for fibers longer than 1 micrometer and 0.1 × 10^6^ fibers/gram of dry tissue, for fibers longer than 5 μm. Both chrysotile and amphiboles were present in the different tissues analyzed.

Pollice et al. [16] reported the concentration of asbestos fibers in 24 samples from 3 cases. The samples were extracted from different body organs and only 2 were pleural tissue. The authors report that two subjects were affected by asbestosis with an occupational history of asbestos exposure. The case number 3 is a control with no occupational asbestos exposure reported. The analytical procedures described in the study do not specify the detection limit, but, as reported in Table 1 of the paper [16], the minimum concentration detected is 0.1 × 10^6^ fibers/gram of dry tissue. Chrysotile was found in the pleural tissue of one case and amphiboles were found in the other case.

Suzuki & Yuen [17] searched for asbestos fibers in 151 cases of pleural mesothelioma, but a quantitative analysis was reported only in 47 samples from 21 cases (as depicted in Table 2 of the paper). In two cases, pleural tissue was not analyzed. The diagnosis of mesothelioma was confirmed by macroscopic appearance, histology, histochemistry, immunocytochemistry, and sometimes, only using electron microscopy. A history of asbestos exposure was supported by the occupations of the 21 subjects. The authors report the detection limit for each sample analyzed; the lowest seems to be 0.02 × 10^6^ fibers/gram of dry tissue. Both types of asbestos fibers were detected.

The 12 studies included in this review comprised 137 cases, in which 142 samples were analyzed. The studies were performed on autopsy samples or biopsies (some studies did not specify if the biopsies were acquired from autopsies or living patients). Most of the studies (eight) analyzed pleural plaques (or fibrosis) [6,8,9,10,11,12,15,17]; normal pleural tissue was analyzed in four studies [7,13,14,16] and pleural mesothelioma tissue was analyzed in three studies [6,12,17]. All the studies are a collection of cases which were available to the authors; therefore, the level of evidence associated with these studies is that of a case series.

Asbestos fibers were detected in 111 samples (78%) and were not detected (or detectable) in 31 samples (22%). Most of the studies reported both chrysotile and amphiboles fibers [6,9,10,11,12,13,15,16,17]; two studies reported only chrysotile asbestos [7,8] and one study (only one patient) reported amphiboles fibers only [14].

Table 3 describes the detection limits of asbestos fibers in the pleura (expressed in 10^6^ fibers per gram of dry tissue) and analytical technology used. When the detection limits were not specified in the study, we listed the lowest concentration of asbestos fibers reported.

As it is shown in Table 1 (diagnoses) and Table 3 (analytical techniques) the studies included in this review are highly heterogeneous, preventing firm conclusions from being derived.

The studies with the highest number of cases and samples were published by Sebastien et al. (29 cases) [7], followed by Paoletti et al. (21 cases) [13].

Of the 137 cases, 48 were diagnosed with pleural mesothelioma (35%), 38 with pleural plaques or fibrosis (28%), 11 with asbestosis (8%), 8 with pleural effusions (6%), 7 with peritoneal mesothelioma (4%), 6 with lung cancer (4%), and 1 with laryngeal cancer (1%). Other diagnoses were unrelated to possible asbestos exposure (other diseases or cancers, accidental or unknown cause of death). As several cases presented more than one diagnosis, we reported only the main diagnosis for each case, according to the following order: mesothelioma, lung cancer, laryngeal cancer, asbestosis, pleural plaque or fibrosis, and pleural effusion.

Table 4 summarizes the range and type of fibers found in pleural tissues (plaque and normal tissue) analyzed in pleural mesothelioma cases. The study by Sebastien et al. [7], not reported in Table 4, included 4 cases of confirmed pleural mesothelioma where parietal pleura was positive for asbestos fibers, but it did not report a numerical value for these cases (Figure 1 of their manuscript) [7]. Out of 45 mesothelioma cases, 34 samples of pleural tissue (plaque and normal tissue) were taken: 25 samples of pleural tissues and 9 samples of pleural plaques. Asbestos fibers were detected in 16/25 samples of pleural tissue (64%), while in only one study [13] 9/25 samples were below detection limit (36%). In the studies with reported quantitative measures, as expected, asbestos fibers were detected in 100% of pleural plaques (9 out of 9 samples). The fibers most frequently found were amphiboles, which were present in 38% of the samples of pleural tissue and ranged from 0.1 to 0.5 mfgdt, followed by chrysotile (28% of the samples) with a range from 0.275 to 2.6 mfgdt. Amphiboles were found in all 4 studies, in 16 samples out of 30, and chrysotile was found in 3 out of 4 studies, in 12 samples out of 30.

## 4. Discussion

The studies included in this review were highly heterogeneous; they were not controlled studies but case series, and the methods employed for the determination of asbestos fibers were not standardized (e.g., patient selection, electron microscopy examination method, tissue preparation, and so on) and, in some papers, valuable information regarding the beginning of exposure to asbestos and its duration and latency was not reported.

Despite decades of technological advances in electron microscopy, the impact on the detection limits of asbestos fibers seems not to have been influenced, as the minimal value reported in the studies has not changed significantly. For instance, in the study by Sebastien et al. [7], the detection limit calculated was 0.01 × 10^6^ fibers using TEM, while 21 years later, Suzuki and Yuen [17] reported the value of 0.03 × 10^6^ fibers using a high-resolution analytical electron microscopy.

The studies included in this review reported the number of asbestos fibers weighted for either wet or dry tissue. In order to make the results comparable, we transformed the wet tissue values to dry tissue values, multiplying them by 10, as it is customary. One study reported the values of fibers by “cc of fixed tissue”; for this review, we equated this value to one gram of wet tissue.

However, there are some general considerations which may be derived from the data reported.

In the 12 studies reviewed, 78% (111/142) of the pleural tissue samples analyzed showed any type of asbestos fibers. The values of asbestos fibers detected in the positive samples in the individual studies were distributed in a very large range, from as low as 0.016 million of fibers per gram of dry tissue (mfgdt) in the pleural plaque tissue in one study [9] and up to 240 mfgdt in pleural mesothelioma in another study [17].

Regarding chrysotile fibers in the pleural tissue, the maximum number of fibers found was 5.1 mfgdt [7], while in pleural plaque, the maximum detected fibers were 89.7 mfgdt [12]; these two values differ from the maximum number of fibers found in pleural mesothelioma, which reached a concentration of 240 mfgdt in one sample [17].

Likewise, the minimum concentration of fibers overlaps in the three tissues in terms of magnitude in the different studies. Therefore, it is not possible to distinguish a clear pattern that differentiates one tissue from the other in terms of concentration of asbestos fibers.

Most of the studies reported chrysotile and amphiboles fibers in similar concentration ranges. Chrysotile-only fibers were reported in two studies [7,8] and amphiboles-only fibers were reported in just one patient in only one study [14].

Asbestos fibers of both types have been regularly identified in pleural mesothelioma tissue, although it seems that chrysotile fibers are the prevalent ones [6,12,17]; no sample of pleural mesothelioma tissue was negative for asbestos fibers in any study.

Asbestos fibers were not detected in 22/50 samples (44%) of pleural tissue in two studies [7,13]. In the study by Sebastien et al. [7], 13 cases were negative for asbestos fibers out of 29 cases examined (45%); however, for 7 of these cases, exposure to asbestos was possible, based on the job title, whereas in the remaining 6 cases asbestos exposure was unlikely. In the study by Paoletti et al. [13], 9 cases were negative for asbestos fibers out of 21 examined (43%); nevertheless, for 4 of these cases, an exposure to asbestos is possible based on the job title, whereas in 5 cases, asbestos exposure seems unlikely.

Asbestos fibers were not detected in 9/26 samples (35%) of pleural plaques in 2 studies [9,15]. In the study by Warnock et al. [9], and 7 cases were negative for asbestos fibers out of 12 cases examined (58%). However, for four of these cases an exposure to asbestos is possible based on job title, whereas in the remaining three cases, asbestos exposure seems unlikely. In the study by Boutin et al. [15], 2 cases were negative for asbestos fibers out of 14 examined. However, these two cases were part of six unexposed subjects examined (that is, in all exposed subjects, pleural plaques were positive for asbestos fibers). However, the specific results about pleural plaques negative for asbestos fibers need to be critically considered, since pleural plaques are highly prevalent in workers exposed to asbestos, so one would expect that asbestos fibers should be present in this tissue, if they were induced by asbestos exposure.

The percentage of samples in which asbestos fibers were not detected in samples of pleural tissue (44%) or samples of pleural plaques (35%) contrasts with the results of the analyses of lung tissue, where asbestos fibers (of both types) are regularly found in exposed subjects [19] and in autopsy series of the general population [20].

## 5. Conclusions and Perspectives

There are several theoretical reasons why pleural tissue may produce negative results for asbestos fiber determination:-the analytical result could be a “true negative” because, in some subjects, asbestos fibers may not reach the pleura from the lung (but this is unlikely given that it is well known that pleural mesothelioma, and especially pleural plaques, are frequent in workers with true asbestos exposure; moreover, the studies reviewed show that asbestos fibers have also been detected in some subjects with no known occupational exposure to the mineral);-the analytical result could be a “true negative” because fibers are absent in the plural tissue sampled but may be present in adjacent zones; this is particularly likely if the pleural tissue analyzed is a very small sample and, in this case, the analytical result is a “true negative” of the sample analyzed but it is a “false negative” with regard to the pleura as a whole;-the analytical result could be a “false negative” for various issues (pitfalls in the samples preparation or in the microscopic examination, and so on).

If the pleural tissue sample analyzed is large enough, negative results because of an insufficient sample should normally be avoided; although, asbestos fibers seem to concentrate in some specific substructures of the pleura [15], and these substructures are present all over the parietal pleura. It seems unlikely that asbestos fibers concentrate only in specific macroscopic zones of the pleura, as it is not in agreement with the observation that malignant and benign pleural pathology does not seem to be localized only in a very specific area of the parietal pleura.

This scoping review has identified only a few studies, very heterogeneous about representative samples and analytical techniques, with a high risk of inappropriate samples, since, except for autopsy, it is complicated to extract fragments of pleural tissue because of the considerable technical difficulties [21,22].

The presence of asbestos in the pleural tissue of 48 subjects with pleural mesothelioma suggests that the asbestos fiber quantification could be useful to distinguish a mesothelioma related to asbestos exposure from mesotheliomas not associated with the mineral.

Asbestos fibers reach the pleura from the lung and here they produce a variety of responses (pleural plaques, diffuse pleural thickening, malignant mesothelioma), Whereas the concentration of asbestos fibers in normal pleural tissue can be assumed to reflect the fibers arriving from the lung, the same may not be expected for pathological pleural tissues. Since pleural plaques and diffuse pleural thickening result from the production of fibrous tissue in the normal parietal pleura, the concentration of asbestos fibers in these tissues may well differ from the concentration of asbestos fibers in normal pleura. The same is true for mesothelioma tissue; in this case, the proliferation of cancer cells produces a larger mass of tissue than the normal pleura, so the concentration of asbestos fibers in mesothelioma tissue may again differ from the one present in normal pleura.

Thus, we recommend that every time enough mesothelial tissue is available from patients diagnosed with malignant mesothelioma who undergo surgical procedures (for example, pneumonectomy), the measurement of asbestos fibers both in pathological and in normal pleural tissues should become a more frequent, and possibly a routine procedure. This, however, may only be feasible in centers where the technology and resources are available.

Likewise, the electron microscopy technology used in the selected studies has a relevant implication in the quantification of asbestos fibers. As of today, because of technological advancement, asbestos fibers may be recognized better than 50 years ago [22], even using automated techniques [23]. Using electron microscopy technology increases the chances of fiber detection in a good autopsy specimen [21,22]. However, improvement in technology is not enough if it is not paralleled by the use of systematic techniques by suitably trained laboratory personnel.

Besides a systematic study of asbestos in pleural tissues, systematic reviews on quantification of asbestos fibers in the lung parenchyma and a comparison with experimental studies carried out on animal models could provide additional value in the future; we are currently planning a systematic review of studies on asbestos fibers in the lung, asbestos fibers in the lung compared to fibers in the pleura, asbestos fibers in tissues other than pleura or the lung, studies of asbestos fibers in humans vs. animals, and studies of asbestos type and size of fibers found in humans.

Whenever pleural tissue and pleural plaques of human subjects are available, we recommend to systematically search for asbestos fibers and to collect accurate information on occupational and/or environmental exposure. A systematic search for asbestos fibers could fill the knowledge gap about the mineral in normal or pathological pleural tissues, contributing to a better understanding of the development of pleural diseases induced by asbestos.

## Figures and Tables

**Table 1 life-12-00296-t001:** Description of the patients and diagnoses in the included studies.

Reference	N° of Cases	Type of Diagnosis
Le Bouffant, 1974 [6]	5 biopsies	3 pleural mesotheliomas 2 pleural plaques
Sebastien et al., 1980 [7]	29 biopsies	4 pleural mesotheliomas 18 pleural fibrosis 6 pleural effusions 1 lung cancer
Churg, 1982 [8]	4 autopsies	4 pleural fibrosis
Warnock et al., 1982 [9]	12 autopsies (excluding subjects with mesothelioma or lung fibrosis)	2 accidental death 1 lung cancer 4 other cancers 5 other causes
Dodson et al., 1990 [10]	8 autopsies	4 asbestosis 1 laryngeal cancer 3 other causes
Gibbs et al., 1991 [11]	12 autopsies	12 pleural fibrosis
Kohyama & Suzuki, 1991 [12]	10 biopsies	2 pleural mesotheliomas 3 peritoneal mesotheliomas 3 asbestosis 2 asbestosis and lung cancer
Paoletti et al., 1993 [13]	21 biopsies	21 pleural mesotheliomas
Tossavainen et al., 1994 [14]	1 autopsy	1 other cause
Boutin et al., 1996 [15]	14 biopsies	3 pleural mesotheliomas 2 pleural plaques 4 lung cancers 2 pleural effusions 3 other causes
Pollice et al., 1997 [16]	2 autopsies	2 asbestosis
Suzuki & Yuen, 2001 [17]	19 biopsies	15 pleural mesotheliomas 4 peritoneal mesotheliomas

**Table 2 life-12-00296-t002:** Asbestos fibers found in analyzed samples (for additional details the readers are referred to the synthesis of the studies in the text).

Reference	N° of Subjects/n° of Samples°	Type of Tissue Analyzed	Asbestos Exposure	Type of Asbestos Found (n° of Samples with/without Fibers)	N° of Asbestos Fibers (Millions) per Gram of Dry Tissue *		
					Median **	Range	IQ Range
Le Bouffant [6] ***	3/3	Pleural mesothelioma	Occupational	Total (3/0)		1.2–18.6	
	2/4	Pleural plaque, fibrose zone	Occupational	Total (2/0)		3.6–6	
		Pleural plaque, calcified zone	Occupational	Total (2/0)		30–40	
Sebastien et al. [7]	29/29	Pleural tissue	Occupational/unknown	Chrysotile **** (16/13)	Only summary data, median was not reported and cannot be computed	Not detected–5.1	
Churg [8]	4/4	Pleural plaque	Unknown	Chrysotile (4/0)		0.03–1.48	
Warnock et al. [9]	12/12	Pleural plaque	Occupational/unknown *****	Chrysotile (4/8)		0.016–0.058	
				Amphiboles (3/9)		0.021–0.047	
Dodson et al. [10]	8/8	Pleural plaque	Occupational	Chrysotile (8/0)		3.9–21	
				Amphiboles (7/1)		0.16–2.9	
Gibbs et al. [11]	12/12	Pleural fibrosis	Occupational	Chrysotile (12/0)	5.97	1.8–16.1	2.21–11.55
				Amphiboles (12/0)		0.01–0.9	0.055–0.425
Kohyama & Suzuki [12]	10/11	Pleural plaque	Occupational	Chrysotile (10/0)	38	12.1–89.7	29.4–64.3
				Amphiboles (8/2)		0.16–6.81	
		Pleural mesothelioma	Occupational	Chrysotile (1/0)		Only one value (62.1)	
			Occupational	Amphiboles (0/1)			
Paoletti et al. [13]	21/21	Pleural tissue	Occupational/unknown *****	Chrysotile (6/15)		0.275–2.6	
		Pleural tissue	Occupational/unknown *****	Amphiboles (8/13)		0.1–0.5	
		Pleural tissue	Occupational/unknown *****	Total (12/9)	0.45	0.1–2.6	0.350.85
Tossavainen et al. [14]	1/1	Pleural tissue (visceral)	Occupational	Amphiboles (1/0)		Only one value (145)	
		Pleural tissue (parietal)	Occupational	Amphiboles (1/0)		Only one value (12)	
Boutin et al. [15]	14/14	Pleural plaque (parietal, anthracotic)	Occupational	Chrysotile (2/4)		Not detected–1.2	
			Occupational	Amphiboles (5/1)		Not detected–15.66	
			Occupational and Environmental	Chrysotile (0/1)			
			Occupational and Environmental	Amphiboles (1/0)		Only one value (1.28)	
			Environmental	Chrysotile (0/0)			
			Environmental	Amphiboles (1/0)		Only one value (2.42)	
		Pleural plaque (parietal, anthracotic)	Unexposed	Chrysotile (0/6)			
				Amphiboles (4/6)		Not detected–0.6	
Pollice et al. [16]	2/2	Pleural tissue	Occupational	Chrysotile (1/1)		Only one value (0.6)	
				Amphiboles (1/1)		Only one value (0.1)	
Suzuki & Yuen [17]	19/20	Pleural plaque	Occupational	Chrysotile (5/0)		12.1–39.2	
				Amphiboles (5/0)		0.6–1.29	
		Pleural mesothelioma	Occupational	Chrysotile (12/0)	12.3	0.06–240	0.5–56.7
				Amphiboles (4/8)		0.04–0.7	
		Pleural plaque and pleural mesothelioma (mixed)	Occupational	Chrysotile (2/0)		16.6–228.2	
				Amphiboles (1/1)		Only one value (1.8)	
		Pleural plaque and peritoneal mesothelioma (mixed)	Occupational	Chrysotile (1/0)		Only one value (17)	
				Amphiboles (0/1)			

* When original data were reported for wet tissue, the results were multiplied by 10 to convert them to dry tissue. ** Median of an interquartile (IQ) range was calculated only for cell containing 10 values or more. *** The paper does not specify which of the cases were exposed to asbestos or not. **** The paper reports “when a pleural sample was positive for asbestos, almost all of the fibers encountered were of the chrysotile type”. ***** Neither the occupation nor the exposure is well defined.

**Table 3 life-12-00296-t003:** Detection limit of asbestos fibers in the pleura, expressed in number of fibers × 10^6^.

	Detection Limit for Amphiboles (10^6^/gram of Dry Tissue)	Detection Limit for Chrysotile (10^6^/gram of Dry Tissue)	Technology Used
Le Bouffant [6]	1.2 *	1.2 *	Electron microscope and electron X-ray diffraction
Sebastien et al. [7]	Not available	0.01	TEM
Warnock et al. [8]	0.02 *	0.02 *	STEM and energy dispersive X-ray spectroscopy
Churg [9]	Not available	0.03 *	Electron optical microscope, electron diffraction, and energy dispersive X-ray spectroscopy
Dodson et al. [10]	0.16 *	3.9 *	STEM analytical electron with energy dispersive X-ray analyzer
Gibbs et al. [11]	0.01	1.8	TEM and energy dispersive X-ray spectroscopy
Kohyama & Suzuki [12]	0.16	0.16	TEM with a fluorescence screen with a light microscope
Paoletti et al. [13]	0.13 *	0.13 *	TEM and energy dispersive X-ray spectroscopy EDAX
Tossavainen et al. [14]	0.1	No information provided	SEM
Boutin et al. [15]	0.17 *	0.4 *	TEM and energy dispersive X-ray spectroscopy EDAX PV9900
Pollice et al. [16]	0.1	0.6	TEM and energy dispersive X-ray spectroscopy EDAX 9900
Suzuki & Yuen [17]	0.03	0.03	High resolution analytical electron microscope with energy dispersive X-ray spectrometry

* When the DL was not specified, the lowest value observed is reported. TEM = Transmission electron microscope. STEM = Scanning-transmission electron microscope. SEM = Scanning electron microscope.

**Table 4 life-12-00296-t004:** Range and type of fibers found in pleural tissues (plaque and normal tissue) analyzed in pleural mesothelioma cases.

References	Type of Tissue	N° of Samples with/without Fibers	Type of Asbestos Found	Range of Asbestos Fibers (10^6^/gram of Dry Tissue)
Kohyama & Suzuki [12]	Pleural Plaque	2/0	Amphiboles	0.60–1.29
		2/0	Chrysotile	12.1–39.2
Paoletti et al. [13]	Pleural Tissue	8/13	Amphiboles	0.1–0.5
		6/15	Chrysotile	0.275–2.6
Boutin et al. [15]	Pleural Plaque	3/0	Amphiboles	1.28–8.28
		0/3	Chrysotile	
Suzuki & Yuen [17]	Pleural Plaque	2/0	Amphiboles	0.6–1.29
		2/0	Chrysotile	12.1–39.2
	Pleural Plaque/Pleural Mesothelioma (mixed)	1/1	Amphiboles	Only one value (1.8)
		2/0	Chrysotile	16.6–228.2

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
