# Peer review of "Quantitative Assessment of Asbestos Fibers in Normal and Pathological Pleural Tissue—A Scoping Review"

_life, 2022, doi:10.3390/life12020296_

Round 1
Reviewer 1 Report
The manuscript by Carabella-Arias et al. entitled 'Asbestos fibers in pleural tissue--a scoping review' is an interesting and comprehensive review of the limited number of studies which have quantitatively analyzed pleural tissue for asbestos content. In the discussion (lines 375-380), the authors recommend that mesothelioma samples should be routinely analyzed for asbestos fiber content. I have several issues with this suggestion. Firstly, tumors begin with a single cell which divides until billions of cells are present at clinical presentation. Therefore, any fibers found in tumor tissue got there after the fact. Secondly, a very limited number of laboratories in the world have capabilities to perform such analyses. Thirdly, what accumulates in the lungs drives what gets to the pleura, so lung parenchyma is, when available, the favored tissue for fiber analysis. These issues need to be considered by the authors before making such a sweeping recommendation.
Another issue concerns fiber dimensions. This information is not provided in the review and is not available in many of the studies. In the study by Boutin oncogenic fibers were identified in black spots in the parietal pleura (i.e., fibers with dimensions known to be related to disease). On the other hand, most of the chrysotile fibers identified by Suzuki were a micron or less in length (i.e., dimensions not demonstrated to be carcinogenic). The dimensions of fibers identified in pleural tissues need to be addressed, and this may be an area for future studies.
Other minor points are as follows:
1) Material and Methods, line 51, 'undifined' should be 'undefined'
2) In Table 2, for the Gibbs study, the average fiber concentration is reported as 1.79 but the range is reported as 0.01 to 0.9. Please check these numbers.
3) In Table 2, line 88, 'Divided by 10' is ambiguous, since results per gram dry are typically 10 times higher than results per gram wet. (In the text, the authors explain this correctly).
4) In Results, line 204, '5 m micrometer' should be '5 micrometer'
5) Results, line 225, 'were got' should be 'were gotten'
6) In Discussion, line 365, 'plural' should be 'pleural'
7) References, No. 22, more details need to be provided (e.g., only initials of the authors are presented)
8) References, No. 23, authors and title are duplicated
Reviewer 2 Report
Asbestos fibers in pleural tissue - a scoping review
The paper summarizes results from studies on the asbestos fibres in the pleura in humans. It includes results from samples of both normal and pathological pleural tissue, the latter including plaques and mesothelioma.
It is a valuable piece of work. It will be useful as reference for the other researchers active in the field. However it must be improved to deserve publication.
As far as I know, it includes all the papers with human data of interest for the project.
The Conflict of interest section is not complete. Prof Boffetta and Prof Violante are known for their involvement in litigation cases on asbestos related diseases issues. This must be included.
General comment
The review content should be improved in some aspects before deserving publication.
Some of the studies also present data on the concentration of fibers in the lung. This is a relevant information that must be included in the review, whenever possible also with presentation of the correlation between values in pleural and lung samples.
Information in the original papers often includes the total concentration of fibers (all asbestos types) or the distribution by length and diameter, total or by asbestos type. This valuable information was not presented in the review and I strongly suggest to add it.
There is no mention of animal studies, neither in experimental nor in the observational setting. For instance Dumortier et al (Occup Environ Med 2002;59:643–646) measured asbestos fibers in the lung and pleura of wild goats living in an asbestos contaminated area and observed asbestos concentration greater in the pleura than in the lungs.
There is no reason to exclude the peritoneal mesothelioma cases, if fiber measurements were taken in the pleura (and in the lung!).
Results
The key information is presented in table 2.
The tables must be improved in the layout. Now they are very difficult to read. Moreover, the separation between studies should be more clear, possibly drawing separation lines. For table 2 I suggest to put the information by fiber type into columns and to use the rows for type of tissue sample and exposure condition.
Table 2 presents for each fiber type the mean and the range of values in samples with fibers counted, while the number of samples without fibers is presented only for all asbestos types together. This is confusing for the reader and I suggest to add for each fiber type also the number of samples without fibers of the specific type. I suggest the use not the range but a better variability index whenever possible (see below).
The range is a poor measure of variability, that should be used only for very small statistical samples. In some instances the statistical samples number 10 subjects or more and in those instances better variability indexes should be included, e.g. the quartiles. For the very small statistical samples, I suggest to list all the values.
Regarding the studies:
LeBouffant (1974) study presented the number of fibers with no asbestos type specification, therefore the wording ‘Asbestos’ should be used instead of chrysotile+amphiboles. It should also be specified that the four samples on pleural plaque tissue (2 hyaline, 2 calcified) were from the same 2 subjects.
Sebastien et al (1980) and elsewhere. The data in the paper present only the range without more detailed information. I suggest to change the wording ‘No data available’ in ‘Only summary data, average was not reported and cannot be computed’.
Warnock et al (1982) observed 4 subjects with chrysotile and 3 subjects with amphiboles (2 with both) leading to 5 subjects with fibers in pleural plaque tissue. This is not clear from the table.
Id. A comment on the different occupational exposure (see table 1 in Warnock et al.) of subjects with and without detected fibers in needed, here or later in the discussion.
Gibbs et al (1991) presented the number of crocidolite and amosite fibers separately in the same samples. How did you compute the number of ‘Amphiboles’? The mean value (1.79) is greater than the upper range (0.01 - 0.9).
In the Tossavainen et al (1994) study only 1 subject provided data, the wording ‘No data avaliable’ in the ‘Range’ column should be substituted by n=1 (or similar).
Boutin et al (1996) also provided fiber concentration in the antracotic parietal spots and in the normal pleura, in exposed and non-exposed subjects (see their table 2). These results may be considered in the present review. Three patients with pleural mesothelioma presented higher fiber concentration in the pleura than in the lung.
In the Pollice et al (1997) study 2 exposed subjects were analyzed: one showed chrysotile fibers (but no amphiboles) and the other showed amphiboles (but no chrysotile). A further subject, not exposed, showed no fibers. The information in table 2 should be clarified and completed.
Suzuki et al (2001) provided asbestos fiber count for more cases than presented in table 2 of present review (see pag 153 in Suzuki et al). They also provided the proportion of cases with asbestos fibers detected (page 152). Shouldn’t those data be included in table 2 or in the text? The reason for not including the data on fibers in pleural plaques from cases of peritoneal mesothelioma is not clear and should be explained. (Peritoneal mesothelioma cases with asbestos measured in the pleura were included when presenting Kohyama et al’s results).
The text presenting the studies (lines 95 – 222) is correct but sometimes it does not explain the selection criteria that were applied by the authors for the analyses of fiber burden (e.g. Tossavainen provided numerical data for 1 subject only). This leads to numbers that are different from table 2 and may cause difficulty in the reader.
The statements on detection limits (lines 252 – 258) and that on wet/dry tissue (lines 259 – 264) should be moved to the discussion section.
The computation of the ranges (minimum – maximum ) (lines 265 – 278) after pooling the results is statistically nonsense and these statements should be cancelled. The relevant information is already in table 2, for each study. The proper procedure, if it were applicable, should be to compute a meta analytical estimate of a central position statistic (mean or median most often) and to compute its variance. If different groups of subject are lumped together, the range will almost inevitably increase and the variability will be overestimated.
Table 4 should be completed including all the mesothelioma cases, that is adding the cases with sample of mesothelioma tissue and the peritoneal mesothelioma as well.
Please note that the range in the Sebastien et al (1980) study is wrong. The authors should explain how the numerical values were obtained for Sebastien et al (1980) study.
In the Kohyama study were included 7 mesothelioma cases, of which 2 pleural and 5 peritoneal. Only 3 peritoneal cases with pleural plaque fiber measurement were considered here (see table 1), the remaining 2 had fibers measured in the tumour. You should either include them in table 4 (preferred) or report and comment in the text the results.
Tossavainen et al (1994) did not indicate who was the subject presented in their Table 2. Most likely (based on the correspondence between table 2 and table 1 he was not affected by mesothelioma but he died with asbestosis. Please check and confirm or amend.
Suzuki et al (2001) included 21 mesothelioma cases (15 pleura, 6 peritoneum) but, contrary to the selection for Kohyama et al, the peritoneal were excluded, even if fibers in pleural plaques were measured in two. Fibers in the tumor were also measured for all (pleural and peritoneal mesothelioma cases) and should be reported. As in comment about Kohyama.
Discussion
Line 332 Specify if negative samples were from normal or pathological pleural tissue. Paoletti provided interpretation for the negative results, his comments should be reported.
The interpretation of positive and negative results in pathological samples (plaques and mesothelioma) should take into consideration also the pathological process and the progressive increase of lesion volume. The interpretation given in lines 349 -352 is rather limited.
Boutin et al (1996) suggested that fibres are concentrated in specific areas of the pleura and are not evenly distributed. I did not find any point in the discussion related to the possible negative results because of the sample location in the pleura.
Statement “Thus, we recommend that every time that mesothelial tissue is available from patients diagnosed with malignant mesothelioma, the measurement of asbestos fibers must be part of the diagnostic routine” (line 378 – 380) is not clear. I guess you want to suggest that measurement of asbestos fibers should become more frequent and, possibly, a routine. However, this cannot be part of the diagnostic process because adequate samples would be difficult to obtain and likely the sampling process would harm the patient, with no diagnostic advantage. At the current status of knowledge, the only recommendation ethically and clinically acceptable is to collect pleural samples during surgery, whenever possible, or at autopsy, for research (and not clinical) purpose.
Reviewer 3 Report
The manuscript of Caraballo-Arias and colleagues reviewed the relevance of asbestos fibers in the different pleural diseases such as pleural mesothelioma, pleural fibrosis, lung cancer and others. Authors found that asbestos was detected in 78% of all the pleural samples investigated, and in 22% below the detection limit. They assert that the reviewed literatures lack standardization of procedures and techniques causing false or true negative results. Authors recommend a systematic analysis of pleural samples of subjects, which have been exposed to asbestos to “to fill the gap about the presence of asbestos fibers in normal or pathological pleural tissue.”
The data presented support the conclusions in the abstract. However, some concerns should addressed to improve the impact of the manuscript. Principally, authors failed to put emphasis on the clinical relevance of collected data in the asbestos-associated diseases.
Specific comments
Title- the title is too general and does not include the major findings presented in the manuscript. The data presented demonstrate the presence and relevance of asbestos in the different asbestos-related disorders/malignancy; hence, indicating its critical role in the etiology of these diseases. Authors may wish to amend the title to “Impact of asbestos fibers in pleural diseases – a scoping review” or as appropriate.
Abstract – the main objective of this review should be explicitly stated relative to line 8-10. “Pleural mesothelioma is a rare (..) to asbestos. Studies (..) are scarce.”
Line 20-24, “to fill the knowledge gap about the (..) or pathological pleural tissue”. What is the clinical relevance of this premise relative to the different pleural diseases such as diagnosis. Kindly specify in a broader sense.
Introduction
Line 44-46, kindly give some reasons how this review will advance the field of the identification asbestos in normal or pathological pleural tissues.
Line 31, “pleural mesothelioma is a (..) in people not exposed to asbestos” – this is not correct.
Under this section, pleural mesothelioma, pleural plaque, asbestosis being a major components of this manuscript should be introduced briefly such as the implication of asbestos in these diseases.
Authors should be consistent in using the term pleural mesothelioma; either pleural mesothelioma or malignant mesothelioma as in lines 29- 43.
Tables
Table 2 contains three similar parameters in Table 1 such as reference, No of patients and asbestos exposure. For better overview, both Table 1 and Table2 may be fused together in a landscape format.
Line 96-222, the descriptions of all the studies summarized in Table 1 and Table 2 are too detailed – summarizing these 12 studies as in line 223-241 are sufficient enough to highlight the principal findings.
Line 96-222, “reported/studied/assessed the concentration of asbestos fibers” in this section is is redundant. Kindly amend as appropriate.
Discussion
It is an accepted notion that mesothelioma is considered an occupational disease attributed to prolonged exposure to asbestos. Although the reviewed studies are highly heterogeneous, non-controlled and methods were not standardized etc. (line 303-308), the reviewed studies showed that 78% of the analyzed pleural samples contained asbestos. These results support the current consensus that asbestos is the most possible cause of pleural mesothelioma and other pleural diseases attributed to occupational or environmental exposure. Considering this, subjects exposed to asbestos should take precautionary measures to avoid these detrimental health problems. An appropriate revision is highly suggested.
That chrysolite and amphibole in the reviewed literatures have been regularly identified in pleural mesothelioma merit discussion compared to current literatures indicating other types of asbestos.
Line 332-340; line 341-348, as the details of the studies have been vividly described previously and summarized in Tables 1 and 2, these sections should be shortened to indicate the major findings.
A separate section for Conclusions and Perspectives such as in line 375-397 is highly recommended. Authors should highlight that asbestos is an occupational hazard causing asbestos-driven diseases/malignancy such as pleural mesothelioma, and equally important, affirming that asbestos exposure is an established causative factor in the pathogenesis of pleural mesothelioma.
Line 395-395, “to fill the knowledge gap about the presence (..) pleural tissue.” – the implied idea warrants a broader clinical view.
Round 2
Reviewer 1 Report
No further comments.
Author Response
We thank the reviewer for the previous suggestions about our manuscript, which we think have given us the opportunity to enhancing the article for its readability and impact.
Yohama Caraballo-Arias
Reviewer 2 Report
I completely disagree on the new statement in line 34: "Asbestos ... is the primary risk factor for pleural mesothelioma(V.T. et al., 2018)" as ASBESTOS IS THE PRIMARY RISK FACTOR FOR MESOTHELIOMA IN PLEURA, PERITONEUM AND OTHER MESOTHELIAL SITES. The statement in the first version was better and the paper must return to it.
Line 288: The statement "The most abundant type of fiber found was amphiboles" is not very clear as it is not evident if it refers to the proportion of positive samples (correct, as amphibole fibres were was found in 16/30 and chrysotile in 12/30) or to the concentration of fibres (not correct, as amphibole concentration was 0.1 to 0.5 mfgdt, chrysotile concentration was 0.275 to 2.6 mfgdt). Please make it clearer.
Line 329: not clear : "was 2.6 or 5.1 mfgdt". Should be 5.1?
Line 382-384 The statement "- the analytical result could be a “true negative” because fibers are absent in the plural tissue sampled but may be present in adjacent zones; this is particularly likely if the pleural tissue analyzed is a very small sample)" is correct if the primary outcome is the sample result but it is a ‘false negative’ result if the expected outcome is the estimation of presence and characteristics of asbestos fibres in the pleura. Please expand or discuss.
In the Conflict-of-interest section there are different details provided by the two authors PB and FSV about their conflicts. The disagreement should be harmonized, and also Prof Boffetta should state for what parts in the trial he provided the expert witness.
The authors should make clear in the conclusions that they have in preparation papers on the analysis of the relation between fibres concentration in pleura vs lung, of the fibre size and of animal vs human studies, according to my previous comment.
Author Response
Please see attachement

Reviewer 3 Report
Caraballo Arias and colleagues have addressed the issues appropriately.
Author Response

(The authors gave the same response as above.)
